# Data curation via joint example selection further accelerates multimodal learning

**Talfan Evans**[1,*]   **Nikhil Parthasarathy**[1,*]   **Hamza Merzić**[1,2]   **Olivier J. Hénaff**[1,*]

[1]Google DeepMind    [2]University College London
London, UK

## Abstract

Data curation is an essential component of large-scale pretraining. In this work, we demonstrate that jointly prioritizing *batches* of data is more effective for learning than selecting examples independently. Multimodal contrastive objectives expose the dependencies between data and thus naturally yield criteria for measuring the *joint learnability* of a batch. We derive a simple and tractable algorithm for selecting such batches, which significantly accelerate training beyond individually-prioritized data points. As performance improves by selecting from large super-batches, we also leverage recent advances in model approximation to reduce the computational overhead of scoring. As a result, our approach—multimodal contrastive learning with joint example selection (JEST)—surpasses state-of-the-art pretraining methods with up to $13\times$ fewer iterations and $10\times$ less computation. Essential to the performance of JEST is the ability to steer the data selection process towards the distribution of smaller, well-curated datasets via pretrained reference models, exposing data curation as a new dimension for neural scaling laws.

## 1   Introduction

Data quality is an essential driver of performance for large-scale pretraining. Whether in language [17], vision [13], or multimodal modeling [1, 20, 30], training on well-curated datasets has consistently demonstrated that strong performance can be achieved with significantly less data. However, current data pipelines rely heavily on manual curation, which is difficult and expensive to scale. In contrast, model-based data curation [29, 31], which uses features of the model being trained to select high quality data, holds promise for improving the slow, power-law scaling of large-scale pretraining across modalities, both in theory [45] and in practice [13].

Existing methods apply curation at the level of individual data points [10, 40]. Yet the quality of a batch is also a function of its composition, in addition to the summed quality of the data points considered independently. In computer vision, hard negatives (i.e. clusters of points which lie close to one another but contain different labels) have been found to provide a more effective learning signal than trivially solvable ones [5, 19, 32, 39, 43, 48, 51]. In this work we seek to generalize this notion by asking whether model-based data-selection criteria applied to batches of data can accelerate learning beyond what is possible by selecting examples independently.

In multimodal learning, contrastive objectives directly expose the interactions between examples in a batch. We therefore derive a simple and tractable algorithm for joint example selection (JEST), which efficiently selects relevant 'sub-batches' of data from much larger 'super-batches' given their model-based scores. When scoring batches with a pretrained reference model (i.e. *easy-reference*), JEST accelerates learning relative to uniform batch selection, significantly improving independent example selection with the same model (as in CLIPScore [20]). When scoring batches according to their *learnability*, which also takes into account the learner's loss [31], JEST improves further, matching the performance of state-of-the-art models [52] with up to $13\times$ fewer training iterations.

38th Conference on Neural Information Processing Systems (NeurIPS 2024) Track on Datasets and Benchmarks.

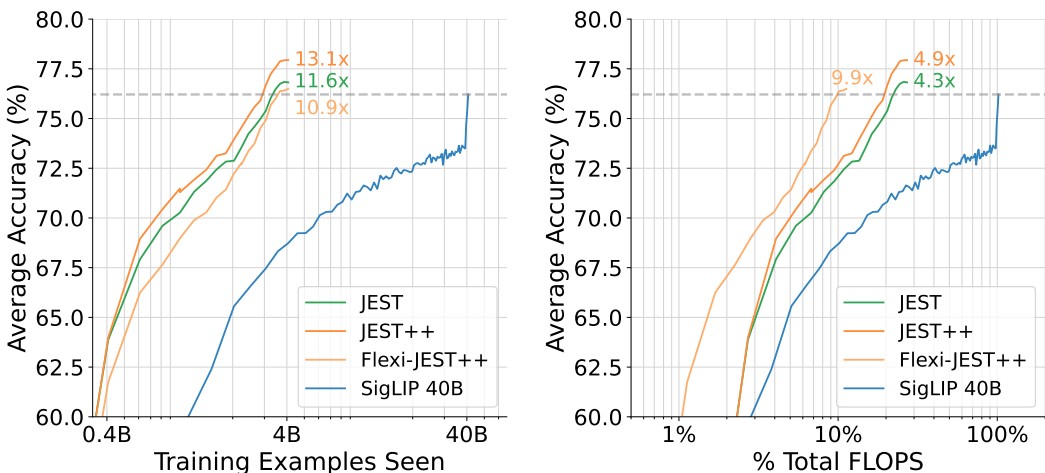

Figure 1: **Joint Example Selection accelerates multimodal pretraining.** Our JEST / JEST++ methods bootstrap from two small, strongly curated datasets (WebLI-curated / WebLI-curated++) to actively curate web-scale datasets. Flexi-JEST++ uses variable patch sizing to reduce the cost of curation. **Left**: Training with JEST matches the performance of the uniform 40B SigLIP baseline with up to 13× fewer iterations. **Right**: Even when accounting for the cost of scoring, our best variant is almost 10× more FLOP efficient.

Discovering highly learnable batches requires sifting through much larger super-batches of raw data. We make learnability scoring of large batches tractable by leveraging recent advances in online model approximation which reduce computation while still providing useful predictions [4, 27, 53]. By training a single model at multiple resolutions in parallel, we efficiently apply the model for scoring large super-batches, find their most learnable sub-batch, and spend more valuable computation for learning on them. Thanks to savings in both learning and example scoring, we reduce the overhead of scoring from 133% to 10% additional FLOPs while maintaining significant gains in training efficiency. This approximate scoring framework (Flexi-JEST) produces state-of-the-art models with 11× fewer iterations *and* 10× fewer FLOPs.

Finally, we find that central to the performance of our framework is the ability to steer the curation process towards the distribution of smaller, well-curated datasets. This occurs naturally with the model-based selection criteria we consider through the concept of a pretrained reference model, which prioritizes examples that most resemble the data it was trained on. Crucially, we find this process to enable strong *data quality bootstrapping*: a reference model trained on a small curated dataset can effectively guide the curation of a much larger dataset, allowing the training of a model which strongly surpasses the quality of the reference model on many downstream tasks.

## 2 Related Work

**Offline curation: example-level data pruning.** Methods for collecting and filtering noisy image-text data initially focused on the quality of the textual captions [22, 8], and proximity to high-quality reference datasets [50, 14]. Instead, model-based filtering approaches use pretrained models (such as CLIP [36] and BLIP [25]) evaluation metrics for curating data via image-text alignment [14, 20, 30]. Critically, all of these methods are applied *independently across examples*, which fails to account for the relevance of dependencies across examples in a batch.

**Offline curation: cluster-level data pruning.** Other methods such as semantic redundancy reduction [1, 2, 45] or core-set selection [6, 18] have proposed to curate based on the marginal importance of data points given other data points in their vicinity. However these methods are based on a heuristic that is decoupled from the training objective. In contrast, our method enables joint-example selection that is specifically tailored to accelerating the contrastive pretraining objective function.

**Online data curation with model-based scoring.** Pre-filtering using the curation procedures described above can lead to large increases in data quality. However, fixed curation strategies do not

take into account that the relevance of a training example can change over the course of learning, limiting their utility at scale [16]. These concerns are addressed by online data curation methods [13, 28, 29, 31], which identify high-quality examples *not yet learned by the model*. Our work generalizes these by applying model-based criteria to batch-level (rather than example-level) losses, and selecting data accordingly.

**Hard negative mining.** A long literature has described the efficiency gains afforded by choosing the right set of negative examples in classical metric-learning [5, 19, 32, 43, 48, 51] as well as modern contrastive learning [39, 47]. We generalize hard negative mining in two ways: 1) we jointly mine for both positive and negative pairs 2) we explore prioritizing *learnable* negatives, which are hard for the learner but easy for a pretrained model.

**Model approximation.** Several works have demonstrated that smaller models can be used as proxies for much larger models [10, 49, 13] for data selection. However, several techniques have recently been developed that allow inference-time trade-offs between computation and performance, allowing smaller models to be "embedded" without the need for separate training. For Vision Transformers [12], dropping patches [27] or layers [53], or reducing token resolution [4] produce characteristic trade-offs [26]. This work is the first to use these techniques in the context of online data selection.

# 3 Methods

## 3.1 Model-based batch-selection criteria

We refer to the model which we are interested in training as the *learner*. Assuming we have a "super-batch" $\mathcal{D}$ (of size $B$) examples to learn from, we wish to extract a sub-batch $\mathcal{B} = \{\boldsymbol{x}_i, i \in [1, ..., b]\} \subset \mathcal{D}$ that is maximally relevant for learning. Prioritized sampling [29, 41] performs this by scoring individual examples, then sampling in proportion to these scores. In this work we instead score entire sub-batches, and sample according to these batch-level scores. We consider model-based scoring functions, which use the losses from the learner model and/or pre-trained *reference* models.

**Hard learner.** An intuitive heuristic would be to prioritize batches $\mathcal{B}$ that have a high loss under the learner with parameters $\theta$: $s^{\text{hard}}(\mathcal{B}|\theta) = \ell(\mathcal{B}|\theta)$, which has the desirable property of discarding trivial data. This heuristic has been proven to work for small, clean datasets [34, 45] but tends to do more harm than good for larger, less curated datasets [13] since it will also up-sample noisy data.

**Easy reference.** In contrast, one could also choose to up-sample data that is "easy" (has low loss) for a pre-trained *reference* model with parameters $\theta^*$: $s^{\text{easy}}(\mathcal{B}|\theta^*) = -\ell(\mathcal{B}|\theta^*)$. This *easy reference* heuristic has been used successfully in multi-modal learning to identify high-quality examples [20, 42], but does not reflect the current state of the learner and can therefore be overly dependent on the choice of reference model [13] and not scale to large compute budgets [16].

**Learnability.** Finally, Mindermann et al. [31] propose to combine these scores, prioritizing with the difference of losses: $s^{\text{learn}}(\mathcal{B}|\theta, \theta^*) = s^{\text{hard}}(\mathcal{B}|\theta) + s^{\text{easy}}(\mathcal{B}|\theta^*) = \ell(\mathcal{B}|\theta) - \ell(\mathcal{B}|\theta^*)$. This heuristic, which we refer to as *learnability* scoring throughout, has the advantage of up-sampling data that is both unlearned and learnable, and has been shown to accelerate large-scale learning even when prioritizing individual examples in isolation [13]. In this work, we therefore mainly consider *learnability* scoring but for completeness also provide ablations with *easy reference* scoring.

The ratio of the "sub-batch" and "super-batch" sizes define the *filtering ratio* $f = 1 - b/B$, i.e. the proportion of data discarded at each iteration. For a given learner batch size $b$, higher filtering ratios increase the cost of scoring as they require more inference passes on the super-batch.

## 3.2 Joint example selection (JEST) for multimodal learning

**Multimodal learning losses.** Given the availability of internet-scale datasets of paired images and text, multimodal learning has become the default means of training visual representations. Contrastive learning aims to maximize the alignment of these two modalities for paired examples, while minimizing the alignment of unpaired examples. Both sigmoid- [52] and softmax-contrastive [36] losses achieve this with a batch-level loss $\ell(\mathcal{B}|\theta) = \frac{1}{b}\sum_{i=1}^{b} \ell(\boldsymbol{x}_i|\theta, \mathcal{B})$, where the conditional loss $\ell(\boldsymbol{x}_i|\theta, \mathcal{B})$ can use a sigmoid or softmax contrast function (see Equations 1 and 2). Since

---

Algorithm 1: Joint example selection: sigmoid loss

---

```python
def jointly_sample_batch(learner_loss, ref_loss, n_chunks=16, filter_ratio=0.8, method="learnability"):
  scores = learner_loss - ref_loss if method == "learnability" else - ref_loss
  n_images = scores.shape[0]                      # scores.shape = [B, B]
  n_draws = int(n_images * (1 - filter_ratio) / n_chunks)  # Size of each chunk.
  logits_ii = np.diag(scores)                     # Self-similarity scores.
  inds = random.choice(logits_ii, n_draws)        # Sample first chunk.

  for _ in range(n_chunks - 1):
    is_sampled = np.eye(n_images)[inds].sum(axis=0) # Binary indicator of current samples [n_images,].
    logits_ij = (scores * is_sampled.reshape(n_images, 1)).sum(axis=0) # Negative terms ij [n_images,].
    logits_ji = (scores * is_sampled.reshape(1, n_images)).sum(axis=1) # Negative terms ji [n_images,].
    logits = logits_ii + logits_ij + logits_ji    # Conditional learnability given past samples.
    logits = logits - is_sampled * 1e8            # Avoid sampling with replacement.
    new_inds = random.choice(n_images, n_draws, p=np.exp(logits))
    inds = np.concatenate((inds, new_inds))       # Expand the array of indices sampled.
  return inds                                     # Gather and return subset indices.
```

---

Zhai et al. [52] demonstrate the sigmoid-contrastive loss to be a more scalable alternative to the softmax-contrastive one, we adopt it by default.

**Joint example selection.** Because the contrastive loss of a batch decomposes into a sum of conditional losses, the *joint learnability* of the batch $s(\mathcal{B}|\theta, \theta^*) \triangleq \ell(\mathcal{B}|\theta) - \ell(\mathcal{B}|\theta^*) = \frac{1}{b}\sum_{i=1}^{b} \ell(\boldsymbol{x}_i|\theta, \mathcal{B}) - \ell(\boldsymbol{x}_i|\theta^*, \mathcal{B}) = \frac{1}{b}\sum_{i=1}^{b} s(\boldsymbol{x}|\theta, \theta^*, \mathcal{B})$ also decomposes into a sum of *conditional learnabilities* $s(\boldsymbol{x}|\theta, \theta^*, \mathcal{B})$ of each example given other examples in the batch. We wish to sample batches in proportion to their joint learnability, i.e. $p(\{X_k\} = \mathcal{B}) \propto \exp(s(\mathcal{B}|\theta, \theta^*))$, which is enabled by a sequential approach inspired by blocked Gibbs sampling (see Algorithm 1). Given a subset of examples $\mathcal{B}_n$ already included in the batch at iteration $n$, we compute the *conditional learnability* of remaining candidate examples $\boldsymbol{x}_i$ with $s(\boldsymbol{x}_i|\theta, \theta^*, \mathcal{B}_n)$, and sample a new chunk of examples $\{X_k\}$ independently without replacement according to these probabilities: $p(X_k = \boldsymbol{x}_i) \propto \exp(s(\boldsymbol{x}_i|\theta, \theta^*, \mathcal{B}_n))$.

We update the batch by appending this chunk to the previous subset: $\mathcal{B}_{n+1} = \mathcal{B}_n \cup \{X_k\}$, and iterate until $n = N$, the number of chunks. The first chunk $\mathcal{B}_1$ is sampled using unconditional learnability (i.e. self-similarity only) $s(\boldsymbol{x}_i|\theta, \theta^*, \varnothing) = \ell(\boldsymbol{x}_i|\theta, \varnothing) - \ell(\boldsymbol{x}_i|\theta^*, \varnothing)$ where the unconditional losses are computed as $\ell(\boldsymbol{x}_i|\theta, \varnothing) = -\alpha \boldsymbol{z}_i^{\text{im.}} \cdot \boldsymbol{z}_i^{\text{txt}}$ for the softmax-contrastive loss and $\ell(\boldsymbol{x}_i|\theta, \varnothing) = \log\left[1 + \exp(-\alpha \boldsymbol{z}_i^{\text{im}} \cdot \boldsymbol{z}_i^{\text{txt}} + \beta)\right]$ for the sigmoid-contrastive loss. We find that a relatively small number of chunks ($N = 16$, sampling $b/N = 2{,}048$ examples independently at each iteration) is sufficient to recover batches with very high learnability (see section 4.1).

## 3.3 Efficient scoring and multi-resolution training

**Efficient scoring with online model approximation.** Scoring large super-batches increases the cost per iteration, lowering the efficiency gains in terms of total FLOPs. While [13] required additional small, proxy models to efficiently score data on behalf of larger learners, we remove this requirement by using online model approximation. We only approximate the image encoding since this accounts for the bulk of the cost of each inference pass [27]. For this we adopt the FlexiViT architecture [4], which lowers the image resolution while minimally degrading performance (see Appendix Figure A.4 for a comparison to patch dropping [12, 27]). In our experiments, we evaluate the super-batch with 32×32-pixel patches , which gives a 72% reduction in FLOPs and 67% reduction in wall-clock time vs. full-resolution scoring at patch size 16×16 [27] (see Section A.3).

**Multi-resolution training.** While we want to score examples at low resolution (i.e. with large patches), at test-time we wish to evaluate the model at full resolution (i.e. with small patches). To enable both resolutions of computation, we simply train at both resolutions. Specifically, given a sub-batch $\mathcal{B}$ for learning, we randomly split it into two halves, $\mathcal{B}^{lo}$ and $\mathcal{B}^{hi}$, and encode each half with a different resolution: $\mathcal{Z}^{lo} = \{f^{\text{im}}(\boldsymbol{x}; \theta, p = 32), \boldsymbol{x} \in \mathcal{B}^{lo}\}, \mathcal{Z}^{hi} = \{f^{\text{im}}(\boldsymbol{x}; \theta, p = 16), \boldsymbol{x} \in \mathcal{B}^{hi}\}$. These images embeddings are then concatenated together as $\mathcal{Z} = \mathcal{Z}^{lo} \cup \mathcal{Z}^{hi}$ and the rest of training proceeds as usual. In addition to allowing for efficient scoring, multi-resolution training itself yields a gain in efficiency: since $\mathcal{B}^{lo}$ is processed with $4\times$ fewer tokens, it also benefits from close to a $4\times$ speed-up. If $\mathcal{B}^{lo}$ and $\mathcal{B}^{hi}$ each account for half of the batch, the cost of multi-resolution training on $\mathcal{B}$ is 64% of the FLOPs and 67% of the time of full-resolution training. Pseudocode for the full-resolution JEST and multi-resolution Flexi-JEST implementation is detailed in Algorithm A.1.

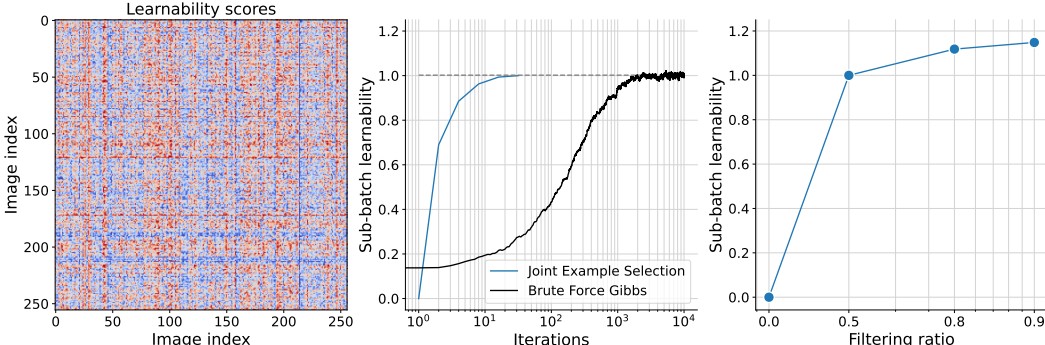

Figure 2: **Joint example selection yields more learnable batches. Left:** the learnability of a batch is highly structured and non-diagonal. **Middle:** Joint example selection quickly discovers sub-batches with high learnability, on-par with brute-force Gibbs sampling. **Right:** the learnability of sampled batches improves with higher filtering ratios (i.e. selecting from larger super-batches).

**Training datasets.** We train the learner model in all JEST experiments on the WebLI dataset [9], specifically a billion-scale subset of English image-text pairs loosely filtered with image-text alignment [52], using the `big_vision` codebase [3]. To train reference models, we use smaller high-quality datasets. JEST/Flexi-JEST reference models are trained on a strongly filtered 100M scale subset of WebLI filtered for high text and image quality and image-text alignment, which we refer to as "WebLI-curated". We additionally explore *scaling data curation* (JEST++/FlexiJEST++) using reference models trained on "WebLI-curated++" which adds approximately 600M additional web-scraped image-text pairs filtered with the same strong curation pipeline.

## 4    Experiments

### 4.1    Joint example selection yields learnable batches

We start by evaluating the efficacy of joint example selection (JEST) for selecting learnable batches. To gain an intuition for our method, we start by visualizing the learnability matrix (i.e. the difference in loss between learner and reference models, for all pairs of examples in the batch). JEST is designed to sample sub-matrices of examples in proportion to their summed learnability. Since the matrix is strongly non-diagonal (Figure 2, left), independent selection will clearly be sub-optimal.

With a small number of iterations (corresponding populating the batch with $N = 16$ chunks), we find the learnability of the sub-batch to quickly increase, matching the learnability of batches extracted by brute-force Gibbs sampling requiring thousands of iterations (Figure 2, middle).

For filtering ratios of 0.5, 0.8, and 0.9, we select sub-batches of 32,768 examples from super-batches of size 65,536, 163,840 and 327,680 respectively. In Figure 2, right, we find that the learnability of the sub-batch increases with larger filtering ratios. In summary, our joint example selection (JEST) algorithm is an effective and efficient means of selecting learnable batches during training.

### 4.2    Joint example selection accelerates multimodal learning

We now investigate the effect of training on more learnable batches, as selected by our JEST algorithm. All runs use a reference model trained on WebLI-curated, a ViT-B/16 and Bert-B image-text dual encoder, 3 billion training examples, and the sigmoid-contrastive loss. Figure 3 (left) shows the average performance on multiple downstream tasks (ImageNet 0-Shot/10-Shot accuracy and COCO image-to-text/text-to-image retrieval) over the course of training. We find that JEST significantly accelerates learning, reaching the final performance of the 3B-uniform baseline after only 2B, 1B, and 0.67B training examples, when using filtering ratios of 50%, 80%, and 90% respectively. At larger filtering ratios we observe similar training instabilities to those observed for larger batch sizes [52], necessitating a modification to stabilize the Adam optimizer ($\beta_2 = 0.95$) and suggesting that data curation with JEST can be thought of as increasing the effective batch size (Appendix A.1, A.2).

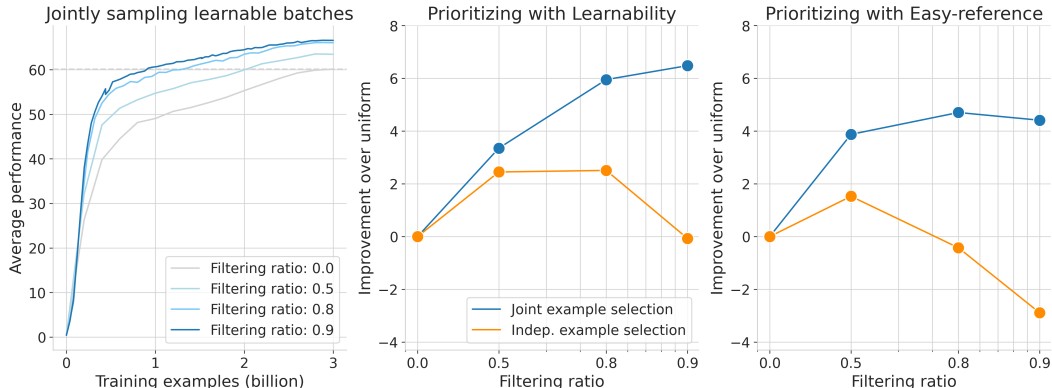

Figure 3: **Joint example selection accelerates multimodal learning. Left:** training on the most learnable sub-batch selected from super-batches that are 2×, 5×, or 10× larger significantly accelerates multimodal learning. **Middle:** Jointly prioritizing *learnable batches* yields significantly better results than simply prioritizing individual examples. **Right:** joint examples selection also improves *easy reference* prioritization, although *learnability* scales better with more aggressive filtering.

In terms of final performance, JEST also delivers significant gains of up to 6% when filtering 90% of data (Figure 3, middle, blue curve). Notably, this scaling behavior is absent from previous selection methods based on independent prioritization of individual examples (Figure 3, middle, orange curve). Finally, we assess whether JEST also improves prioritization criteria other than learnability. Figure 3, right, shows the performance of models with *easy-reference* prioritization, for varying filtering ratios. Consitent with learnability-based prioritization, JEST strongly outperforms independent example selection, particularly for high filtering ratios (where independent example selection leads to a regression in performance). Prioritising data with the highest loss produced smaller gains and degrades more quickly as we filter more data (Appendix Figure A.6). Since learnability-based JEST yielded the best scaling behavior we retain this criterion for subsequent experiments.

### 4.3 Synergies between multi-resolution training and online batch selection

Joint example selection with *learnability* scores becomes more efficient as larger fractions of each batch are filtered. However, the cost of scoring results in a significant overhead: filtering 80% of the super-batch results in 4× more FLOPs per iteration than IID training, or 2.3× when caching the reference-model scores (Appendix A.3). Although JEST is significantly more efficient in terms of training iterations (hereinafter 'training efficiency'), the additional scoring FLOPs reduce its compute efficiency relative to the IID baseline (Figure 1, left vs. right). We therefore also investigated a compute efficient variant, Flexi-JEST, which uses multi-resolution training and low-resolution scoring to reduce the total overhead to only 10% vs. the baseline (Figure 4, left; see Section A.3).

What is the effect of these approximations on performance? As might be expected, the per-iteration performance of Flexi-JEST decreases relative to the JEST, although still produces significant speed-ups over IID (Figure 1, left; Figure 4, middle). However, the decrease in per-iteration performance is more than favorable when accounting for the decrease in total FLOPS: our best Flexi-JEST model produces the same average performance as a 40B Siglip run with 9.9× fewer FLOPs, and 2× fewer than full-resolution JEST (Figure 1, right; Figure 4, middle).

What is the relative contribution of efficient scoring and multi-resolution training in Flexi-JEST? We conducted an ablation where we varied the fraction of the selected batch trained at full and low resolution (i.e. the relative sizes of $\mathcal{B}^{hi}$ and $\mathcal{B}^{lo}$; see Methods). For fair comparison we ensure the learner spends the same FLOPs by increasing the number of training iterations as we send more data to the approximate model, since the FLOPs per iteration decrease in this case (see Section A.6 for details). Figure 4 (right) shows that the IID baseline performance increases with larger fractions of data sent to the approximate model, consistent with a growing literature on the FLOP-efficiency of approximate training [4, 27, 11, 38]. Nevertheless, Flexi-JEST significantly outperforms the multi-resolution baseline as soon as it trains the approximate model (e.g. with as little as 25% data). These experiments demonstrate a synergy between multi-resolution training and joint example selection, as the former yields efficient and accurate scoring capabilities for accelerating the latter.

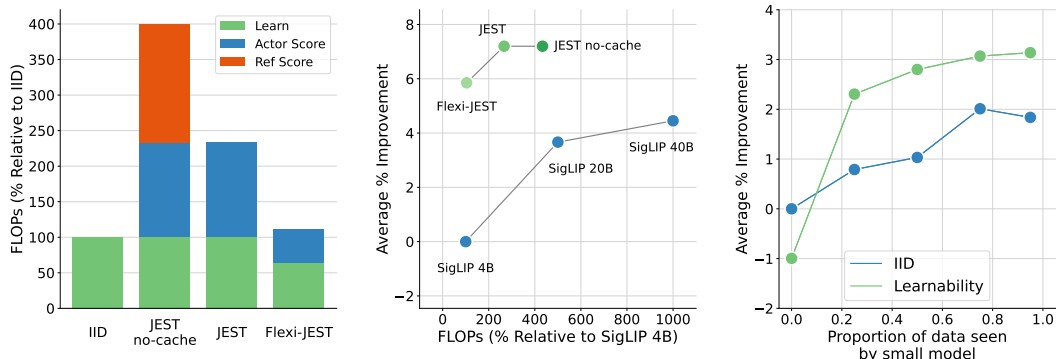

Figure 4: **Efficient scoring and multi-resolution training** . **Left:** In scoring large super-batches with the learner and reference models, JEST incurs a large computational cost per iteration. By caching the fixed reference model scores in the dataset, this overhead can be cut in half. Efficient scoring and multi-resolution training further reduce this to be comparable to standard IID training. **Middle:** Flexi-JEST improves the total FLOP-efficiency of JEST over standard IID training. **Right:** Multi-resolution training improves FlexiJEST more than standard IID training. Without multi-resolution training (left-most point) Flexi-JEST underperforms the IID baseline (due to an untrained approximate model), but quickly improves with even a small amount of co-training (25%).

Our results also point to a pareto front of data curation strategies. If maximizing training speed or training efficiency is desirable at the expense of computation, the full-resolution JEST method produces up to a $13\times$ speed up relative to a comparable IID training run. If FLOPs should be minimized at the expense of training efficiency, Flexi-JEST produces the most favorable trade-off. We note that the scoring of the next batch can be implemented on separate devices, in parallel with training, potentially further reducing the additional wall-clock time.

### 4.4    Joint example selection enables strong data-quality bootstrapping

At the heart of learnability-based scoring is a reference model trained on a small, curated dataset of our choosing. How does JEST performance vary as a function of different curation strategies that trade off quality vs. quantity? Furthermore, do improvements in JEST training correlate with the performance of the reference models or are these metrics decoupled?

**Understanding quality vs. quantity trade-offs**. We explore three scales of curation, each being a subset of the original WebLI dataset: *weak* (billion-scale) curation with image-text alignment (ITA) filters, *moderate* (300M scale) curation with either ITA filters or text-quality (TQ) filters, and *strong* (100M scale) curation with a combination of TQ, ITA, and additional image-quality (aesthetic) filters. Throughout, we refer to this strongly curated subset as "WebLI-curated".

We train standard SigLIP encoders on these four WebLI subsets for 10 epochs each, and use them as reference models for JEST training on the full WebLI dataset. Across curation methods, reference model performance and JEST performance appear to be decoupled (or even anti-correlated; Figure 5, left), consistent with previous findings for fixed data curation [14]. Whereas increasing curation (and decreasing dataset size) yields weaker models, when used as reference models for JEST pretraining they have the opposite effect: JEST with a strongly-curated reference benefits from a 2.7% improvement, moderate a 1.5% improvement, and weak a 0.3% improvement.

**Scaling data curation**.  We hypothesized that the general decoupling between reference model performance and JEST performance might simply be explained by the dataset size limits imposed by data curation. To understand this effect, we trained 5 reference models on WebLI-curated while varying the total examples seen (250M to 3B). In this context, Figure 5 (right) shows a striking correlation between improved reference models and better JEST pretraining. This suggests that the "decoupling" phenomenon can be mostly attributed to the saturation of the reference model as a result of the reductions in dataset size following curation.

We note that the correlation in Figure 5 (right) starts to break down when the dataset is saturated, i.e. after 10 epochs or 1B examples seen. We therefore demonstrate how scaling data curation benefits JEST by augmenting WebLI-curated to a total of approximately 600M examples sourced from an

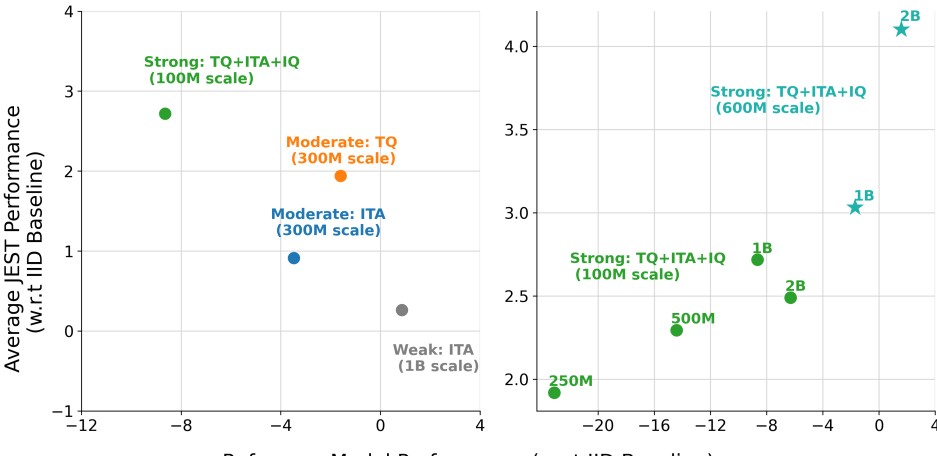

Figure 5: **Scaling strong data curation improves JEST performance. Left:** We compare JEST performance vs. reference model performance (relative to the uniform baseline) for 4 curation types: 'weak' curation with image-text alignment (ITA), 'moderate' curation with ITA or text-quality (TQ), and 'strong' curation (using a combination of TQ, ITA, and additional image-quality (IQ). **Right:** We use our best reference dataset (TQ+ITA+IQ) and evaluate JEST vs. reference performance varying the number of examples seen during reference pretraining. There is a strong correlation between additional reference training and JEST performance that saturates after 1B examples seen. By scaling strong data curation to a 600M dataset, this saturation is broken as both reference model and JEST performance improve for the 1B *and* 2B reference training.

expanded set of image-text pairs. Critically, these examples all still pass the same set of strong TQ, ITA, and IQ filters—we denote this dataset as "WebLI-curated++". We find that this scaled dataset allows us to break the 2B saturation point for "WebLI-curated" as both reference model and JEST performance (Figure 5, Right: ⋆) improves significantly. We therefore use WebLI-curated++ for our best models, JEST++ and FlexiJEST++.

## 4.5 Comparison to prior art

We now compare to prior art, including the state-of-art SigLIP model trained for 40 billion examples [52] as well as recent strong CLIP variants. Table 1 shows that our most training-efficient model, JEST++, sets a new state-of-the-art on both ImageNet and COCO all while using 10× fewer iterations and 4× less compute. On COCO in particular, JEST++ improves the previous state of the art by over 5%. Our most compute-efficient model, Flexi-JEST++, also surpasses the previous SoTA on average, while using 9× less compute. Training JEST for longer furthered these gains (see Appendix Table 3).

Our results also scale gracefully with model size. Training with a ViT-L learner and ViT-L reference trained on the same WebLI-curated++ dataset, JEST++ continues to yield strongly accelerated learning, matching the SigLIP ViT-L 40B baseline with only 4B examples seen (Table 1, bottom).

Finally, we apply JEST++ for pretraining on two publicly available datasets: LAION-2B [42] and DataComp-1B [15]. The DataComp experiments compare against Data Filtering Networks (DFN) [14] and are shown in Appendix Table 5. We find that JEST++ strongly surpasses DFN, and verify that these gains come primarily from switching from offline, independent selection to online joint-example selection using learnability scoring. For the LAION experiments, we follow the standard practice of removing unsafe image-text pairs [44], but do not otherwise pre-filter the dataset. JEST++ strongly surpasses previous methods for offline data curation, despite using 4× fewer training examples than the previous state-of-the-art (Table 2). With this training budget, SigLIP pretraining severely under-performs all methods, further highlighting the benefits of JEST. JEST with a reference model trained on the smaller LAION-400m subset, which is not a curated subset but instead an independent set collected in a similar manner to LAION-2B, did not outperform IID training on the latter.

| Method | Variant | # Train | FLOPs % | | Mean $\Delta$ | ImageNet-1K | | COCO | |
|---|---|---|---|---|---|---|---|---|---|
| | | | Per Iter. | **Total** | | 10-S | ZS | I2T | T2I |
| CLIP [36] | B | 13B | 100 | 32 | $-11.8$ | | 68.3 | 52.4 | 33.1 |
| EVA-CLIP [46] | B | 8B | 100 | 20 | $-4.6$ | | 74.7 | 58.7 | 42.2 |
| OpenCLIP [21] | B | 34B | 100 | 85 | $-5.8$ | | 70.2 | 59.4 | 42.3 |
| LessIsMore [7] | B | 11B | 100 | 28 | $-5.9$ | | 70.8 | 58.3 | 42.5 |
| SILC-S [33] | B | 20B | 380 | 190 | $+0.2$ | 68.9 | 76.6 | 66.2 | 48.7 |
| SigLIP [52] | B | 40B | 100 | 100 | 0.0 | 70.3 | 76.7 | 65.2 | 47.4 |
| JEST++ | B | 4B | 233 | 23 | $+2.8$ | **70.3** | 76.9 | **70.3** | **53.3** |
| Flexi-JEST++ | B | 4B | 110 | **11** | $+0.9$ | 68.2 | 75.8 | 68.0 | 51.2 |
| CLIP [36] | L | 13B | 100 | 32 | $-11.0$ | | 75.5 | 56.3 | 36.5 |
| EVA-CLIP [46] | L | 4B | 100 | 10 | $-3.4$ | | 79.8 | 63.7 | 47.5 |
| OpenCLIP [21] | L | 32B | 100 | 80 | $-6.3$ | | 74.0 | 62.1 | 46.1 |
| SigLIP [52] | L | 40B | 100 | 100 | 0.0 | 77.1 | 80.5 | 69.5 | 51.2 |
| JEST++ | L | 4B | 233 | **23** | $+1.8$ | 75.5 | **80.5** | **71.1** | **54.8** |

Table 1: **Comparison to prior art.** FLOP % are measured relative to SigLIP [52]. Mean denotes the average performance over all metrics. "Per Iter." denotes FLOPs per iteration.

| Method | # Train | IN1K ZS | COCO |
|---|---|---|---|
| LAION-440M [35] | 12.8B | 64.1 | 48.1 |
| SemDeDup [1] | 8.8B | 64.3 | 48.9 |
| DBP [2] | 5.3B | 65.5 | 48.4 |
| DBP [2] | 3.6B | 64.1 | 45.7 |
| SigLIP [52] | 1.3B | 57.2 | 43.3 |
| JEST++ | **1.3B** | **66.8** | **54.8** |

Table 2: **Comparison to LAION pretraining.** JEST++ strongly surpasses prior art while requiring significantly fewer training iterations. COCO performance denotes the average of image-to-text and text-to-image retrieval.

## 5   Discussion

We proposed a method, JEST, for jointly selecting the most learnable batches of data which significantly accelerates large-scale multimodal learning, surpassing the previous state-of-the-art with up to $10\times$ fewer FLOPs and $13\times$ fewer examples. Our experiments point to the strong potential for "data quality bootstrapping", using small curated datasets to guide learning on much larger, uncurated ones.

Recent work has shown that filtering datasets without knowledge of downstream training can ultimately limit performance [16]. Our results further demonstrate that dynamically constructing useful batches improves pretraining efficiency beyond individually selected examples. These findings therefore advocate for *foundation distributions*—either through pre-scored datasets with *easy-reference* JEST, or dynamically adjusted to the demands of the model with *learnability* JEST—as a more general and effective replacement to generic foundation datasets.

**Limitations.** While our method has accelerated multimodal learning of canonical downstream tasks, it has relied on small, well-curated reference datasets which specify the distribution to prioritize within much larger uncurated data. We would therefore encourage future work exploring the inference of reference datasets from the set of downstream tasks of interest.

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

# A    Appendix

## A.1    Optimisation robustness

Our initial experiments used the default ADAM optimizer parameters. At a filtering ratio of 50%, we observed good gains over IID as described previously. Increasing the filtering ratio to 80% however did not produce further gains. Inspecting the training curves showed training instabilities (Figure A.1, Left). In the original SigLIP paper [52], the authors found similar instabilities when increasing the batch size, but found that setting $\beta_2 = 0.95$ (from $\beta_2 = 0.999$) ameliorated the problem.

We find the same behaviour (Figure A.1). Although the filtering ratio does not directly affect the training batch size (which is constant in our experiments), this finding suggests that filtering for salient data has the effect of increasing the *effective* batch size. Although this suggests an importance sampling interpretation, applying a re-weighting to the selected data decreased performance in our experiments. We leave this for further work - it is possible that further performance could be extracted from optimizer tuning at higher filtering ratios.

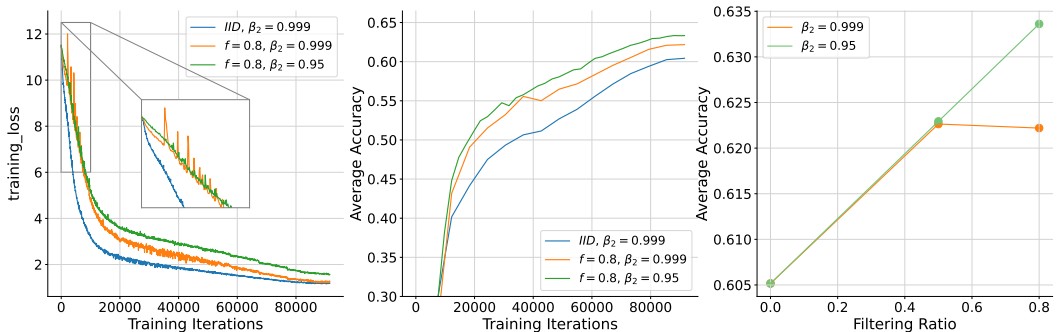

Figure A.1: **Aggressive data prioritization requires robust optimization to perform well. Left:** With standard optimization settings (e.g. $\beta_2 = 0.999$ in the Adam optimizer), aggressive data prioritization leads to instabilities (spikes) in the optimization process. Setting $\beta_2$ to the more stable value of 0.95 remediates this instability. **Middle, Right:** Stable optimization and strong prioritization together yield large improvement gains.

## A.2    Effects of varying the training batch size

It is well known that the performance improvements saturate when increasing the training batch. In [52], increasing the batch size beyond 32K was found to actually decrease performance, even after adjusting the $\beta_2$ parameter. In our experiments, we use 32K batch size throughout.

The observation that filtering larger amounts of data produced the same loss spikes as observed by [52] suggests that the training batches selected by JEST might correspond to a much larger *effective* batch size. To investigate, we conducted an ablation in which we instead fixed the super-batch size and progressively decreased the training batch size (i.e. changing the filtering ratio by decreasing the amount of training data, instead of increasing the size of the super-batch as done throughout the paper).

The results in Figure A.2 demonstrate that, as we decrease the batch size (increase the proportion of data filtered) for a fixed super-batch size of 160K, the performance drops predictably for IID training (Left) but decreases much more slowly for JEST training (Middle / Right). Notably, for a halving of the batch size from 32K (corresponding to our f=80% experiments throughout) to 16K (filtering 90%), there was no noticeable performance drop.

These results suggest that, at 32K training batch size, our experiments might be already operating at close to the optimal *effective* batch size. We did not conduct further ablations, but it is possible that a more favourable FLOP improvement could be achieved by simultaneously increasing the super-batch size and decreasing the training batch size.

These results suggest an importance sampling interpretation of learnability scoring - assuming the "True" mini-batch gradient is given by the expectation of the gradients from IID samples from the

data, JEST is sampling only the data that contributes most to that expectation. This suggests that most of the gradient information can be reconstructed from a small number of data points. Although JEST does not explicitly sample based on the magnitude of the gradients of the data, it was demonstrated previously via a simple Taylor expansion argument that the two are equivalent in the case where learnability scores are small [13].

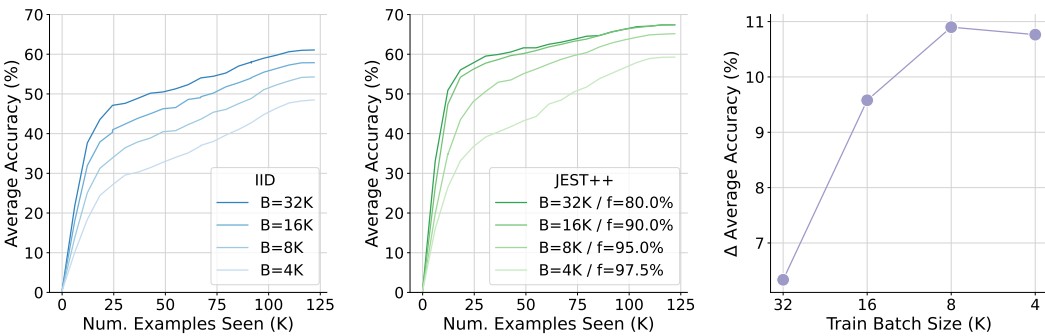

Figure A.2: **Effective Batch Size Experiments** Right: Difference in average performance between JEST++ and IID training as a function of training batch size. Instead of increasing the filtering ratio by increasing the super-batch size as done in previous experiments, we instead fix the super-batch size and reduce the training batch size. There was no noticeable drop from halving the training batch size, suggesting that further efficiency gains might be achieved by training on less data in addition to filtering from a larger pool.

## A.3  FLOP calculations for JEST / Flexi-JEST

We assume that training on a single data point cost approximately $C_{\text{IID}} = 3F$ forward passes $F$ of the learner model [23]. The cost for a single JEST update can therefore be computed as:

$$C_{\text{JEST}} = 3F + FB/b - F = F(2 + B/b)$$

where $B$ and $b$ are the super-batch and sub-batch sizes respectively and $f = 1 - b/B$. The base JEST method does not use approximations on the learner, which allows us to cache the forward pass during scoring and re-use it for the gradient computation. Relative to an IID update, the cost of a single JEST iteration at a filtering ratio $f = 0.8$ comes out as $\alpha_{\text{JEST}} = 7/3 = 2.33$. Flexi-JEST uses two approximations. Firstly, we split the training batch 50:50 between the full:approximate learner, effectively parallelising the method of [27] and reducing the per-data point cost of training. Secondly, we approximate the learner when performing scoring, which reduces the cost of scoring. The overall cost for a single Flexi-JEST update can therefore be computed as:

$$C_{\text{Flexi-JEST}} = 3F(0.5 + 0.5A) + AFB/b$$

where $A$ is the FLOP reduction factor resulting from model approximation (i.e. increasing the patch size, see 3.3). Note that we can no longer cache the forward pass from scoring since it is computed with an approximate version of the learner. Relative to an IID update, the cost of a single Flexi-JEST iteration at a filtering ratio $f = 0.8$ and approximation factor $A = 0.25$ comes out as $\alpha_{\text{Flexi-JEST}} = 1.04$. In practice, [27] estimated the FLOP reduction from a doubling in patch size as closer to $A = 0.28$, which is slightly higher than the $A = 0.25$ expected by reducing the number of patches by $0.5^2$. We use this more conservative calculation ($\alpha_{\text{Flexi-JEST}} = 1.10$) throughout.

## A.4  Caching reference model scores

Since the reference model is pre-trained and fixed, its scores do not vary over the course of a training run and can be cached within the dataset. For independent example selection we only need to store the scalar scores. However, since data is not likely to be sampled from the training set in the same order in which it is initially scored (e.g. if the batch size varies), the batch composition is unknown ahead of training, which will affect the computation of the scores.

```python
cfg = ConfigDict(
    n_chunks=16,
    filter_ratio=0.8,
    method="learnability",
    method="jest", # or "flexi-jest"
    softmax_score_gain=100.0,
)

def sigmoid_nll(params, embeds):
  zimg, ztxt = embeds
  logits = np.dot(zimg, ztxt.T) # [B, B]
  logits = logits * params["alpha"] + params["beta"]
  eye = np.eye(zimg.shape[0])
  m1_diag1 = -np.ones_like(logits) + 2 * eye

  nll_mat = -log_sigmoid(m1_diag1 * logits)
  nll = np.sum(nll_mat, axis=-1).mean()

  return nll, nll_mat # [,], [B, B]

def get_scores_sigmoid(embeds, embeds_ref):
  _, nll_mod = sigmoid_nll(params, embeds) # [B, B]
  _, nll_ref = sigmoid_nll(params, embeds_ref) # [B, B]
  if cfg.scoring == "learnability":
    scores = nll_mod - nll_ref
  elif cfg.scoring == "easy_ref":
    scores = - nll_ref
  return scores * cfg.softmax_score_gain

def loss_fn(params, batch):
  images, texts = batch
  approx = True if cfg.method == "flexi-jest" else False

  # Score and sub-sample the initial super-batch
  embeds = model.forward(images, texts, params, approx=approx) # [5B, D]
  embeds_ref = batch["embeds_ref"] # Pre-cached in dataset
  scores = get_scores_sigmoid(embeds, embeds_ref) # Get scores
  sub_inds = jointly_sample_batch(scores, cfg.n_chunks, cfg.filter_ratio, cfg.learnability)
  images, texts = stop_grad(images[sub_inds]), stop_grad(texts[sub_inds]) # [B, ...]

  # Split batch for co-training
  images_full, images_approx = images[::2], images[1::2] # [B/2, ...], [B/2, ...]
  texts_full, texts_approx = texts[::2], texts[1::2] # [B/2, ...], [B/2, ...]

  # Compute overall loss
  embeds_full = model.forward(images_full, texts_full, params, approx=False) # [B/2, D], [B/2, D]
  embeds_approx = model.forward(images_approx, texts_approx, params, approx=approx) # [B/2, D], [B/2, D]
  zimg = np.concatenate([embeds_full[0], embeds_approx[0]], axis=0)
  ztxt = np.concatenate([embeds_full[1], embeds_approx[1]], axis=0)
  loss, _ = sigmoid_nll(params, (zimg, ztxt))

  return loss
```

To amortize the cost of reference model scoring across training runs for joint example selection, we therefore instead store the *embeddings* from the reference mode. For a ViT-B/16, the embeddings (text and image) are of size 768, which are considerably smaller but not negligible in comparison to the raw data points. Given these embeddings, the super-batch contrastive matrix can be recomputed before sub-sampling to obtain the sub-batch.

## A.5 Contrastive loss ablations

Contrastive learning maximizes the alignment of image and text modalities for paired examples, while minimizing the alignment of unpaired examples, with batch-level losses $\ell(\mathcal{B}|\theta) = \frac{1}{b}\sum_{i=1}^{b}\ell(\boldsymbol{x}_i|\theta, \mathcal{B})$.

Each data point $\boldsymbol{x}_i$ is comprised of an image and associated text which are embedded with their respective encoders as $\boldsymbol{z}_i^{\text{im}} = f^{\text{im}}(\boldsymbol{x}_i; \theta)$ and $\boldsymbol{z}_i^{\text{txt}} = f^{\text{txt}}(\boldsymbol{x}_i; \theta)$.

In softmax-contrastive learning [36], the conditional loss is

$$\ell(\boldsymbol{x}_i|\theta, \mathcal{B}) = -\frac{1}{2}\left(\log\frac{\exp(\alpha\boldsymbol{z}_i^{\text{im}}\cdot\boldsymbol{z}_i^{\text{txt}})}{\sum_j \exp(\alpha\boldsymbol{z}_i^{\text{im}}\cdot\boldsymbol{z}_j^{\text{txt}})} + \log\frac{\exp(\alpha\boldsymbol{z}_i^{\text{im}}\cdot\boldsymbol{z}_i^{\text{txt}})}{\sum_j \exp(\alpha\boldsymbol{z}_i^{\text{txt}}\cdot\boldsymbol{z}_j^{\text{im}})}\right) \quad (1)$$

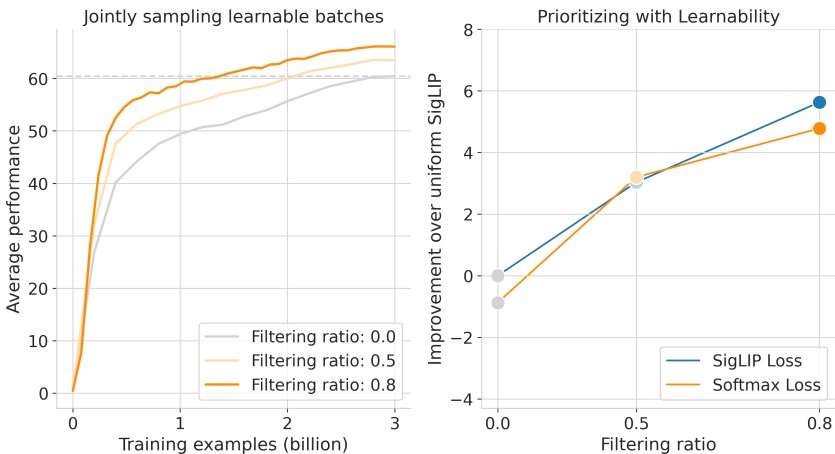

Figure A.3: **JEST is robust to the choice of contrastive loss (softmax vs. sigmoid).** We confirm the robustness of JEST to variations of contrastive learning by testing with the standard softmax-based loss. **Left:** Similar to the SigLIP version, softmax contrastive learning is accelerated by JEST and benefits from larger filtering ratios. **Right:** Compared to the SigLIP loss, the uniform sampling softmax baseline is slightly worse. Despite this at a filtering ratio of 0.5, JEST improves over the baseline softmax by a larger margin, such that the performance gap is closed. However, at a filtering ratio of 0.8, there remains a small gap (similar to the baseline).

whereas in sigmoid-contrastive learning [52], the conditional loss is

$$\ell(\boldsymbol{x}_i|\theta, \mathcal{B}) = \log\left[1 + \exp(-\alpha \boldsymbol{z}_i^{\text{im}} \cdot \boldsymbol{z}_i^{\text{txt}} + \beta)\right] + \sum_{j \neq i} \log\left[1 + \exp(\alpha \boldsymbol{z}_i^{\text{im}} \cdot \boldsymbol{z}_j^{\text{txt}} - \beta)\right]. \qquad (2)$$

Although we leverage the sigmoid pairwise contrastive loss (SigLIP) formulation for our main results, a natural question is whether JEST benefits the standard softmax contrastive learning in Eq. 1. In Fig. A.3, we show that JEST is indeed robust to the choice of contrastive loss. In the right panel, we see that the gains over the baseline softmax are comparable to, if not greater than, the gains for the SigLIP loss. However, due to the degradation in the softmax baseline relative to the baseline SigLIP, the combination of JEST with SigLIP is preferred.

## A.6 Comparing approximation methods

We compared two canonical strategies for online model approximation. Both ablations are conducted at ∼75% FLOP reduction by either dropping 75% of patches or doubling the patch size (see Main Section 3). We vary the proportion of data used for approximate and full-resolution training ($\lambda$ and $1 - \lambda$, respectively), keeping the total number of FLOPs used by the learner the same. Since the cost of one training iteration is proportional $0.25\lambda + 1 - \lambda$, we divide the number of training iterations by this factor to keep the training budget constant with respect to $\lambda$. For $\lambda \in [0.0, 0.25, 0.5, 0.75, 0.95]$, this results in training budgets of $[3, 3.69, 4.8, 6.86, 10.43]$ billion examples seen.

Our results (Figure A.4) demonstrate that downscaling data by decreasing the resolution effects a much more favourable trade-off than dropping a subset of patches. Both methods perform differently out-of-distribution, but only FlexiViT benefits significantly from co-training. We adopt this strategy throughout the paper.

## A.7 Training configuration

Our default training configuration follows that of SigLIP [52], with a ViT-B/16 and Bert-B image-text dual encoder, training on WebLI for 3 billion examples with a batch size of 32k and the sigmoid-contrastive loss. The vision encoder takes images resized to (256 x 256) and the text-encoder tokenizes text with the sentencepiece tokenizer [24] trained on the english C4 dataset [37]. We crop the text to the first 64 tokens. The initial learning rate is 0.001, warmed up linearly during the first 1%

of training, followed by cosine decay. We use a weight decay 0.0001, gradient clipping to a maximum norm of 1.0, and the Adam optimizer with $\beta_2 = 0.95$. We split training across 256 TPUv5e chips.

For the LAION experiments in Table. 2, we use an architecture matched to the prior art (ViT-B/32 vision encoder and resized image inputs to (224 x 224)). Otherwise, we use the same training settings as above. We note that this batch size is comparable to that used in the prior art and we find similar results using the softmax-contrastive loss instead of the sigmoid-contrastive loss.

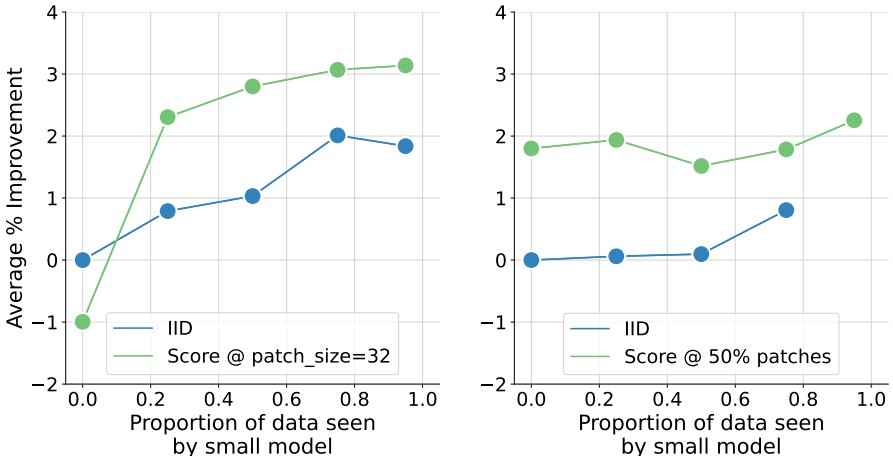

Figure A.4: **Comparing model approximation strategies.** Comparing Flexi-JEST and PatchDrop-JEST at a 50% filtering proportion. On the x-axis, we vary the proportion of data used to co-train the scoring model. Green curve shows JEST runs, blue curves shows equivalent IID run without data curation. All runs conducted at isoFLOP, see Fig. 4. **Left:** FlexiViT scoring performs badly 0-shot out of distribution inference, but quickly recovers with co-training. **Right:** Patch dropping is more robust to 0-shot inference, but doesn't benefit as much from co-training.

| Method | Variant | # Train | FLOPs % | | Mean $\Delta$ | ImageNet-1k | | COCO | |
| | | | Per Iter. | **Total** | | 10-S | ZS | I2T | T2I |
|---|---|---|---|---|---|---|---|---|---|
| SigLIP [52] | B | 40B | 100 | 100 | 0.0 | 70.3 | 76.7 | 65.2 | 47.4 |
| JEST | B | 5B | 233 | 29 | $-0.5$ | 68.2 | 75.5 | 66.1 | 47.9 |
| Flexi-JEST | B | 13B | 110 | 36 | $+1.2$ | 69.4 | 76.6 | 68.2 | 50.2 |
| JEST++ | B | 4B | 233 | 23 | $+2.8$ | 70.3 | 76.9 | 70.3 | 53.3 |
| Flexi-JEST++ | B | 4B | 110 | **11** | $+0.9$ | 68.2 | 75.8 | 68.0 | 51.2 |
| JEST++ | B | 10B | 233 | 58 | $+\mathbf{4.0}$ | **72.3** | **77.6** | **71.8** | **53.9** |
| Flexi-JEST++ | B | 10B | 110 | 28 | $+3.1$ | 71.1 | 77.2 | 70.2 | 53.3 |
| SigLIP [52] | L | 40B | 100 | 100 | 0.0 | 77.1 | 80.5 | 69.5 | 51.2 |
| JEST++ | L | 4B | 233 | **23** | $+1.8$ | 75.5 | 80.4 | **71.1** | **54.8** |

Table 3: **JEST continues to improve with longer training runs.** FLOP % is measured relative to SigLIP [52]. Mean denotes the average performance over all metrics. "Per Iter." denotes FLOPs per iteration. 10B training runs of both JEST++ and FlexiJEST++ continue to improve over the 4B results presented in main Table 1. JEST and Flexi-JEST, which use the WebLi-curated reference dataset both perform strongly on a per-FLOP basis, with Flexi-JEST also outperforming the SigLIP 40B baseilne on average.

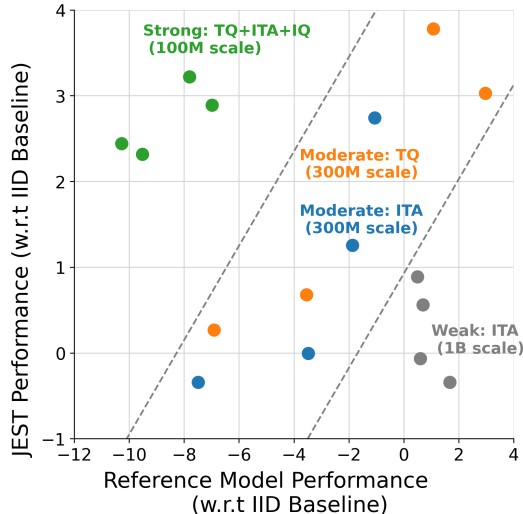

Figure A.5: **Data curation can steer reference models to perform well in specific domains, which is correlated with the specialized performance of corresponding JEST models.** We unpack the averaged results in Fig. 5 (left) by displaying performance across 4 evals (ImageNet-ZS, ImageNet-10-shot, COCO I2T, and COCO T2I). Weakly curated datasets do not show any correlation across evals due to the lack of any strong effect on active learning. However, with moderate curation, there is a strong correlation between reference model and JEST performance across evaluations. This indicates that depending on the type of curation, one can specialize references to certain domains which then *transfers specialization to the learner model*. The correlation is also visible for strongly curated data; however, this type of curation seems to generalize better across domains.

| Method | ImageNet 10-S | 1K | COCO I2T R1 | T2I R1 | Birds 10-S | Caltech 10-S | Cars 10-S | Pets 10-S | Mean |
|---|---|---|---|---|---|---|---|---|---|
| SigLIP (40B) | 70.3 | 76.7 | 65.2 | 47.4 | 75.5 | 92.8 | 92.0 | 91.2 | 76.4 |
| WebLI Curated++ (4B) | 59.6 | 66.5 | 75.4 | 56.9 | 65.6 | 89.9 | 63.0 | 81.9 | 69.9 |
| JEST++ (4B) | 70.3 | 76.9 | 70.3 | 53.3 | 80.4 | 91.6 | 89.5 | 93.1 | 78.2 |

Table 4: **Reference model comparison**. JEST++ using a reference model trained on WebLI curated results in balanced performance across metrics. The reference model itself scores performs well on COCO retrieval, which is inherited by the JEST++ trained model. However, the it scores poorly across other evaluations, whereas the JEST++ model achieves good performance.

| | Ref. Dataset | Train Dataset | ImageNet | ImageNet-Shift |
|---|---|---|---|---|
| DFN (Fang et al. 2023) | HQITP-350M | DC-12.8B | 67.8 | 54.0 |
| Easy-reference, independent selection | Webli-curated++ | DC-1B | 65.0 | 56.9 |
| JEST, easy-reference scoring | Webli-curated++ | DC-1B | 69.3 | 60.4 |
| JEST, learnability scoring | Webli-curated++ | DC-1B | **74.8** | **69.6** |

Table 5: **Ablations with the DataComp dataset.** In [14], the authors use a high-quality reference model trained on a curated dataset of their own (HQITP-350M) to filter the publicly available DataComp 12.8B corpus [15], then demonstrate that training on this subset outperforms training on the publicly available DataComp-1B. Here, we show the results of using our own method for filtering from DataComp-1B. The DFN method is equivalent to our own "easy-reference scoring, independent selection" setting. In the equivalent setting with our own reference model, our method performs similarly to the DFN results. However, adding both learnability based scoring and pairwise selection significantly improves results beyond DFN. We would expect results to improve further by selecting from the larger DataComp-12.8B dataset.

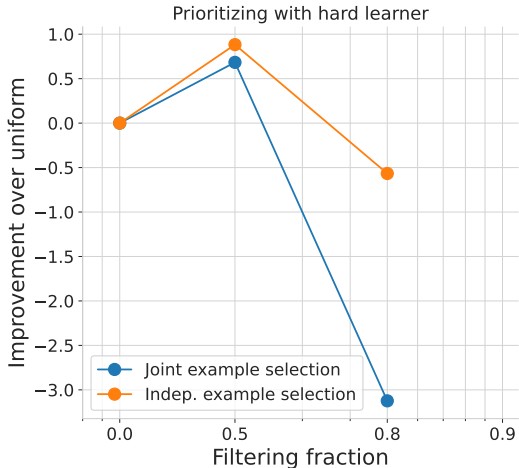

Figure A.6: **Prioritizing data that is difficult for the learner** Compared to *easy-reference* and *learnability* scoring (Figure 2) prioritising data with high learner loss results in small gains at a filtering ratio of 50%, but quickly degrades as we filter larger amounts of data. Joint example selection exacerbates the effect at larger filtering ratios, aligning with the interpretation that *hard-learner* prioritisation prioritises sampling noise in the data.

