# OpenReview forum: "Data curation via joint example selection further accelerates multimodal learning"
_NeurIPS.cc/2024/Datasets_and_Benchmarks_Track — NeurIPS 2024 Track Datasets and Benchmarks Spotlight_

### Official Review · Reviewer_YMd9 · 2024-07-17

**Rating:** 7
**Confidence:** 2
**Correctness:** The evalution of the method and exper…
**Clarity:** The paper is easy to follow.

**Review:**

The idea of this paper is simple but efficient, and potentially profound for large-scale training research. This work has demonstrated the possibility of using small, well-curated datasets to guide the learning process on much larger, uncurated datasets.

**Strengths:**

1. The paper attempts to solve a very important problem, data curation from large batch, and  proposes a novel joint selection schemes can be efficiently implemented.

2. The paper is well-organized, clearly written and easy to understand.

3. The proposed method significantly accelerate the large-scale training with up to 10 times fewer FLOPs and 13 times fewer examples, compared with state-of-the-art SigLIP.

**Additional Feedback:**

My only concern is that this paper does not introduce a new dataset or benchmark but the new approach. I am not sure if this is suitable for the Datasets and Benchmarks Track.

**Documentation:**

There is no public code or data materials included in this submission.

**Ethics:**

There is no ethics concerns.

**Limitations:**

As discussed in the paper, the propose method depends on some small but well well-curated datasets that specify the distribution.

**Opportunities For Improvement:**

Please check limitations.

**Relation To Prior Work:**

The paper has discussed the difference between the existing works including the state-of-the-art SigLIP.

**Summary And Contributions:**

This paper proposes a simple but efficient data curation method referred to as Joint Example Selection (JEST) for large-scale multimodal training. The key idea of JEST is computing joint learnability of samples in batches rather than individual examples leveraging contrastive objectives to expose the dependencies between data. The main contribution of this paper can be concluded into (1) introduce Learnability Scoring to combine hard learner and easy reference, selecting informative examples with both metrics; (2) select examples in batch-level and consider the dependencies  between data; (3) the implementation of this approach is significantly reduce the computation compared with the state-of-the-art data curation methods.

---

> ### Author Rebuttal · Authors · 2024-08-16
>
> **My only concern is that this paper does not introduce a new dataset or benchmark but a new approach. I am not sure if this is suitable for the Datasets and Benchmarks Track.**
>
> We think this work is well suited to the D&B track based on these details cited from the conference submission page (https://neurips.cc/Conferences/2024/CallForDatasetsBenchmarks):
>
> *SCOPE. This track welcomes all work on data-centric machine learning research (DMLR)...This includes but is not limited to: …Data-centric AI methods and tools, e.g. to measure and improve data quality or utility, or studies in data-centric AI that bring important new insight … Advanced practices in data collection and curation that are of general interest even if the data itself cannot be shared.*
>
> We believe that this work is a canonical example of a “study in data-centric AI” /  “Data-Centric AI method”, whose goal is to demonstrate how to “improv(e)...data quality” and “Advance… practices in data collection and curation”.

---

> > ### Comment · Reviewer_YMd9 · 2024-08-22
> > **Response to rebuttal**
> >
> > Dear authors, thank you for addressing all my concerns. I will maintain the score.

---

> > > ### Author Response · Authors · 2024-08-27
> > > **Response to reviewer YMd9**
> > >
> > > We thank the reviewer for acknowledging our rebuttal and confirming that our additional experiments and responses have addressed all of their noted concerns. If there are no additional concerns, we would kindly request that the reviewer consider raising their score. We are happy to answer any further questions.

---

### Official Review · Reviewer_dJ5D · 2024-07-18

**Rating:** 7
**Confidence:** 4
**Correctness:** Yes
**Clarity:** Yes

**Review:**

Pros:
1. I think the results are definitely significant.
2. "Training on a subset of high quality dataset will not improve the model performance but can serve as a good ref model." I think this is a good observation. Although it is a little duplicate with what observed in DFN, but still meaningful.
3. Using multi-resolution to reduce computational overhead is a good contribution.
4. The discussion on two different losses used in JEST (softmaxLoss and SigLIP loss) also gives some insights.


Cons:
1. Use the difference between ref model loss and current model loss is a widely used methods. There are different ways of choosing ref model, but I feel the intuitive explanation behind using high quality is a little confusing. For example, I can understand using a powerful teacher model as ref model, because it encourages the student model to match the performance of teacher model. I can also understand the ref model trained from validation dataset in [13] from their baysian theory perspective. I think “learnability" is a proper word for both cases. But in this paper, although the author claims it is similar to [13], it seems to me closer to the motivation of [14] DFN? That is, the $s^{easy}$ represent the quality and $s^{hard}$ represents the hardness? I hope authors can explain more.
2. I am not very familiar with WebLI dataset.  Is there any reason that the author didn't try other datasets like DataComp many previous works have used? It might be more convincing for a method if it is shown to be effective on different dataset.
3. I am curious that, instead of training a ref model yourself, what if you use existing embedding models like DFN? Maybe those models already suggest high quality examples.
4. It is still intuitively unclear to me why SigLIP is better than softmax Loss.

Overall, I think the result is definitely significant and the methods is practically inspiring. On the other hand, I think the analysis behind this is not that comprehensive. Another minor shortcoming is that there are already lot of ref model related data scheduling paper, which preventing me to give a high score.

**Strengths:**

See pros in review.

**Additional Feedback:**

NA

**Documentation:**

Yes

**Opportunities For Improvement:**

I think the most directly way is to provide a clearer explanation behind their methods.

If they can provide more experimental results on different dataset and ref models that will be better, but I understand it is time-consuming.

**Relation To Prior Work:**

Yes

**Summary And Contributions:**

This paper proposes an online data selection method for multi-modal contrastive pretraining. Compared to many existing methods which employs only one iteration, the paper proposes a multi-iteration schedule. The main technique is to iteratively measure the data helpfulness via computing the difference between its current loss and the ref model's loss within each super-batch.

From the methodology perspectives, there are two main innovations: (1) multi-resolution training (2) use high quality subset to train a ref model.

From practical contribution perspective, it surpasses the state-of-art with up to 13x fewer iterations and 10x less computation.

---

> ### Author Rebuttal · Authors · 2024-08-16
>
> **There are different ways of choosing ref model, but I feel the intuitive explanation behind using high quality is a little confusing. For example, I can understand using a powerful teacher model as ref model, because it encourages the student model to match the performance of teacher model. I can also understand the ref model trained from validation dataset in [13] from their bayesian theory perspective. … But in this paper, although the author claims it is similar to [13], it seems to me closer to the motivation of [14] DFN? That is, the seasy represent the quality and shard represents the hardness? I hope authors can explain more.**
>
> DFN is a “foundation dataset” approach which constructs a static dataset by using reference models trained on small high-quality datasets to filter from the larger and uncurated DataComp-12.8B dataset using image-text alignment.
>
> This is equivalent to the “independent selection” ablation in Figure 3 of our paper. The first difference in our paper is that we use joint example selection to construct good batches, which is only possible with an “online filtering” approach (in contrast to static “foundation dataset” approaches). The second difference is that DFN only uses the reference model for scoring examples, whereas we use learnability scoring.
>
> **I am not very familiar with the WebLI dataset. Is there any reason that the author didn't try other datasets like DataComp many previous works have used? It might be more convincing for a method if it is shown to be effective on a different dataset.**
>
> To alleviate this concern, we additionally ran JEST on the canonical DataComp-1B dataset, see global rebuttal Table A (above). Consistently with our results on WebLI, JEST compares very favorably to other methods in this regime.
>
> I am curious that, instead of training a ref model yourself, what if you use existing embedding models like DFN? Maybe those models already suggest high quality examples.
>
> Yes, if the DFN reference models were available, we could swap them for our Webli-curated / Webli-curated++ trained reference models. Note that DFN uses independent selection and only reference scores (no learner scores; see response to first question above). However, the DFN reference models are not publicly available.
>
> Although we were not able to provide a direct comparison, our new results (see global rebuttal, Table A) show that our reference model filtering DataComp-1B performs similarly to the DFN paper’s induced dataset DFN-2B, which filters from a much larger dataset using the authors’ own high-quality reference dataset (HQITP-350M) and which is not publicly available. We expect that filtering from the original 12.8 DataComp dataset would improve results, as our filtering method allows us to scale dataset size while maintaining data quality.
>
> **It is still intuitively unclear to me why SigLIP is better than softmax Loss.**
>
> Just to be clear, JEST is compatible with both the softmax and SigLIP multimodal contrastive losses, and in Figure A.3 we show that JEST improves both learning objectives similarly. To clarify this compatibility with both learning objectives, we will add an algorithm to the final version detailing how to apply JEST to softmax-contrastive learning.
>
> In Figure A.3 we also see that sigmoid-contrastive learning slightly outperforms softmax-contrastive learning, with or without JEST, consistently with the original SigLIP paper. However we note that the main benefit of SigLIP over softmax-contrastive learning is in its reduced computational and memory footprint, as described in the SigLIP paper.

---

> > ### Author Response · Authors · 2024-08-27
> > **Request of Reviewer dJ5D**
> >
> > We thank the reviewer again for their comments and suggestions and hope that our rebuttal addresses any remaining concerns. We are happy to answer any further questions and would kindly request that the reviewer considers raising their score.

---

> > > ### Comment · Reviewer_dJ5D · 2024-08-28
> > >
> > > Thanks for you clarification. Given additional results and explaination, I will raise my score by 1.

---

### Official Review · Reviewer_fv2x · 2024-07-23
**Good paper, but more experimental evaluations required**

**Rating:** 7
**Confidence:** 4
**Clarity:** The paper is well written and easy to…

**Review:**

The proposed method is simple and well-motivated. In addition, the experiments show that it enables to significantly accelerate training.
However, it is highly recommended to compare the proposed method with more related existing works.

**Strengths:**

(1) The proposed method is simple and well-motivated.

(2) The proposed method provides high performance improvement compared to the existing approaches as well as the IID baseline.

(3) The paper is well written and easy to understand.

**Additional Feedback:**

So far my score is weak 7. I hope to make my score strong 7 or increase it, by the authors addressing the comments in “Limitations” via the rebuttal phase.

**Correctness:**

It seems that the claims made in the submission are reasonable, and the experiments are performed in an appropriate way.
However, I think experimental comparison to more existing works is required.

**Documentation:**

For reproducibility, the authors do not provide the code, but the pseudo code and training details can be found in the paper.

**Ethics:**

It seems that there is no ethical concern with the paper.

**Limitations:**

(1) In the experiments only few of works in the section 2 are compared to the proposed method.
To be more impactful, it seems that the proposed method should be compared with more existing works.

(2) (minor) It would be better that diversity of datasets and models for the experiments increase.

**Opportunities For Improvement:**

Please see “Limitations”.

**Relation To Prior Work:**

The paper discusses how this work differs from previous contributions.

**Summary And Contributions:**

The paper proposed JEST, a joint example selection algorithm based on learnability scoring for multimodal learning.
The key idea is to perform online data curation in batch-level rather than individual examples.
The evaluation shows that the proposed method significantly accelerates multimodal training in comparison with the IID baseline.

---

> ### Author Rebuttal · Authors · 2024-08-16
>
> **(1) In the experiments only few of works in the section 2 are compared to the proposed method. To be more impactful, it seems that the proposed method should be compared with more existing works.**
>
> In order to compare to more prior work, we ran JEST on the DataComp-1B dataset, allowing us to compare to the Data Filtering Networks (Fang et al., 2023) paper, as also suggested by **R3**. (note that we already extensively compare against SemDeDep and DBP, see Table 2 of main paper). In Table A of the global rebuttal (above), we find JEST to compare very favourably, significantly outperforming their method, which uses independent selection and only the reference model for scoring.
>
> **(2) (minor) It would be better that diversity of datasets and models for the experiments increase.**
>
> In addition to the new results on DataComp-1B described above, we remind the reviewer that we already ran JEST on WebLI (Table 1) and LAION (Table 2), with both ViT-B and ViT-L. In all cases JEST strongly surpasses the previous state-of-the-art large-scale pretraining (SigLIP, Table 1) and data-curation methods (DBP, Table 2)

---

> > ### Comment · Reviewer_fv2x · 2024-08-25
> > **-**
> >
> > I appreciate the rebuttal by the authors; the reviewer has read it carefully.
> >
> > Thanks for the new experiments; I will maintain the score.

---

> > > ### Author Response · Authors · 2024-08-27
> > > **Response to reviewer fv2x**
> > >
> > > We thank the reviewer for acknowledging our rebuttal and confirming that our additional experiments and responses have addressed their noted limitations. If there are no additional concerns, we would kindly request that the reviewer consider raising their score. We are happy to answer any further questions.

---

### Official Review · Reviewer_FvDN · 2024-08-04
**Good paper**

**Rating:** 8
**Confidence:** 3
**Correctness:** The claims and method appear correct.…
**Clarity:** This paper is well written.

**Review:**

This paper is good and comprehensive. I have some minor concerns about this work.

- What is the major novelty of this work? Currently, I guess it is a combination of learnability scoring and Multimodal learning losses. Did I miss something?

- Did you consider choosing the probability in another form of the function of learnability scoring? In this work, you chose it to be in proportion to the exponential of learnability scoring. For example, choose the largest learnability scoring, or be in proportion to the square of exponential of learnability scoring.

- In this work, you mainly considered $s^{learn}=s^{hard}+s^{easy}$. You also considered easy reference scoring, i.e., $s^{learn}=s^{easy}$. We might consider more general cases  $s^{learn}=a \times s^{hard}+b \times s^{easy}$. You have already tried the cases (a, b) = (1,1) and (a, b) = (0,1). I would recommend trying (a, b) = (1, 0),  (a, b) = (1, 0.1),  (a, b) = (0.1, 1).

- Suppose your dataset is very clean. How would you expect the accuracy of JEST compared with training on the whole dataset (filtering ratio = 0)?

**Strengths:**

JEST demonstrates a substantial reduction in the number of training iterations required to achieve high performance.

**Additional Feedback:**

N/A

**Documentation:**

This paper showed the main components of its method by a pseudocode.

**Ethics:**

No ethical concerns.

**Limitations:**

The authors discussed the limitations in the Discussion Section.

**Opportunities For Improvement:**

See Review part.

**Relation To Prior Work:**

This paper clearly discussed how this work differs from previous contributions in Related Work Section.

**Summary And Contributions:**

The paper introduces an algorithm named Joint Example Selection Technique (JEST), which enhances multimodal learning by selectively curating training batches that accelerate learning efficiency and reduce computational costs. By leveraging multimodal contrastive learning objectives and a new algorithm for batch selection, JEST prioritizes data batches based on joint learnability.

---

> ### Author Rebuttal · Authors · 2024-08-16
>
> **What is the major novelty of this work? Currently, I guess it is a combination of learnability scoring and Multimodal learning losses. Did I miss something?**
>
> The major novelty of this work is that—in contrast to existing data curation methods, which filter examples independently from one another—we jointly select examples which contain useful dependencies, in addition to being individually more learnable.
>
> As you point out, this lends itself particularly well to multimodal contrastive learning, which exposes those dependencies in the loss itself. We refer the reviewer to the Related Work section for a complete discussion of the relationship between our work and the existing literature.
>
> **Did you consider choosing the probability in another form of the function of learnability scoring? In this work, you chose it to be in proportion to the exponential of learnability scoring. For example, choose the largest learnability scoring, or be in proportion to the square of exponential of learnability scoring.**
>
> Indeed our framework supports all of these choices. In Algorithm 1: `new_inds = random.choice(n_images, n_draws, p=np.exp(inverse_temperature * logits))`. Setting *inverse_temperature = 2* sets the probabilities to be proportional to the square of exponential of learnability scoring as you suggest, and very large inverse temperature (e.g. *inverse_temperature = 1000*) effectively selects examples with the largest learnability scores.
>
> We have ablated the choice of inverse_temperature, finding a monotonic increase in performance (results shown relative to IID baseline) as we decrease temperature (JEST @ *f=0.8*):
>
> | Inverse Temperature | ∆ Mean Accuracy (COCO I2T / T2I, IN-0S) |
> | --- | --- |
> | 1 | +2.6 |
> | 10 | +3.9 |
> | 100 | +4.1 |
>
> We note that the original manuscript omitted inverse_temperature in the equation above. We will fix this in the final version.
>
> **We might consider more general cases slearn = a × shard + b × seasy. You have already tried the cases (a, b) = (1,1) and (a, b) = (0,1). I would recommend trying (a, b) = (1, 0), (a, b) = (1, 0.1), (a, b) = (0.1, 1).**
>
> See global rebuttal, Section 2 and Table B. We have run these experiments with the continuous combinations suggested, but did not find improvements over our best performing learnability scores (*a=1.0*, *b=1.0*). The performance of continuous mixtures interpolates between the performance of the binary mixtures already reported.

---

> > ### Author Response · Authors · 2024-08-27
> > **Request of Reviewer FvDN**
> >
> > We thank the reviewer again for their comments and suggestions and hope that our rebuttal addresses any remaining concerns. We are happy to answer any further questions and would kindly request that the reviewer considers raising their score.

---

> > > ### Comment · Reviewer_FvDN · 2024-08-28
> > >
> > > Thank you for the response. The authors have well addressed my concerns and I do not have other questions. I will increase my rating to 8.

---

### Author Rebuttal · Authors · 2024-08-16

# Summary

We thank the reviewers for the positive scores and thoughtful feedback. To summarise, we propose a method for online data curation that demonstrates “significant” (**R2**) improvements in the FLOP efficiency of pre-training and produces new SOTA results in several image-text modeling tasks.

We are pleased that all reviewers agreed with the significance of the results (**R1** “substantial reductions”, “results look sound”; **R2** “significantly accelerate”, “simple and well motivated”; **R3** “results are definitely significant”; **R4** “potentially profound”, “very important problem”).

We’re also pleased that all reviewers approved of the clarity of the presentation (**R1**, **R2**: “well written”, **R4**:  “well organized” and “easy to follow”) and that the quality of the experimentation was satisfactory (**R1** “comprehensive”; **R4** “evaluation of the method and experiments are well-implemented”).

The main concerns were around adding additional comparisons to other work. In the remainder of this rebuttal we provide additional experiments to address these comments, which we detail and refer further to in the reviewer-specific rebuttals.

# Results on DataComp-1B, comparison to DFN paper

**R2** suggests comparing our results to more existing works (“compare the proposed method with more related existing works” / “(minor) … better that diversity of datasets and models for the experiments increase”). **R3** suggests evaluating performance of our method on another public dataset (“...is there any reason that the author didn't try other datasets like DataComp…”).

**R3** further asks how our embedding model performs relative to the embedding models derived in the Data Filtering Paper (“I am curious that, instead of training a ref model yourself, what if you use existing embedding models like DFN?”).

We ran additional experiments to address these questions. The DFN paper derives a high-quality dataset of 2B examples (DFN-2B)  by filtering the DataComp 12.8B corpus. In their paper, the authors demonstrate that training on DFN2B outperforms training on the publicly available DataComp-1B. We  have access to DataComp-1B but not to DFN-2B / DataComp-12.8. We instead ran further experiments using our high-quality reference model to filter data from DataComp-1B. We are confident that results would improve further if we were to filter data from the larger DataComp-12.8B dataset.

In our additional experiments we trained a ViT-B/16 for 1.28B examples seen and a batch size of 8192, matching the configuration from Table 3 of the DFN paper – the results are shown in the table below. The DFN setup is equivalent to our “Easy-reference scoring, independent selection” setting, which are close to the DFN reported numbers. However, adding JEST sampling (“JEST, easy-reference scoring”) and then learnability scoring (“JEST, learnability scoring”) both add significant performance improvements.

|  | Ref. Dataset | Train Dataset | ImageNet | ImageNet-Shift |
| --- | --- | --- | --- | --- |
| DFN (Fang et al. 2023) | HQITP-350M | DC-12.8B | 67.8 | 54.0 |
| Easy-reference scoring, indep. selection | Webli-curated++ | DC-1B | 65.0 | 56.9 |
| JEST, easy-reference scoring | Webli-curated++ | DC-1B | 69.3 | 60.4 |
| JEST, learnability scoring | Webli-curated++ | DC-1B | 74.8 | 69.6 |
*Table A: Comparison of JEST to prior art on the DataComp dataset*

# Continuous mixtures of reference / learner
**R1** suggests trying non-binary combinations of reference and learner scores for filtering data (“I would recommend trying *(a, b) = (1, 0)*, *(a, b) = (1, 0.1)*, *(a, b) = (0.1, 1)*”). We ran additional experiments at a filtering ratio of *f=0.5* for 4B examples seen, but none outperformed our best  “Learnability” based scoring. Continuous mixtures of reference / learner model coefficients interpolate between the respective settings (i.e. between “Ref. only” / “Learn only” and “Learnability”). Note that differences would be expected to be amplified at *f=0.2* (see Figure 3 of main paper).

| Learner Coeff. | Ref. Coeff. | Name | Mean |
| --- | --- | --- | --- |
| 0 | 0 | IID | 0.609 |
| 0 | 1 | Ref. only | 0.645 |
| 1 | 0 | Learn only | 0.618 |
| 1 | 0.1 | Soft Learn only | 0.634 |
| 0.1 | 1 | Soft Ref. only | 0.645 |
| 1 | 1 | Learnability | 0.650 |
*Table B: Effect of inverse_temperature on JEST performance*.

---

### Decision · Program_Chairs · 2024-09-26

**Decision:**

Accept (Spotlight)

**Comment:**

In this paper, the authors focus on the efficiency of data curation in multimodal learning, then provide an affirmative answer by proposing a simple but effective method, multimodal contrastive learning with joint example selection (dubbed JEST) framework to address this issue. The problem is clear and important, the idea is novel and interesting. Besides, the authors provide a wide of experiments to demonstrate the effectiveness of their method.

I agree with the comments of all reviewers that the contribution of this paper is significant to the multimodal learning community. However, some reviewers raised some minor points in their review which have been addressed in the rebuttal.

Although the paper is good, exploring the reasons behind the success of these techniques and providing intuitive explanations would contribute to the overall scientific contribution of the work.

Therefore, the recommendation is to "Accept" the paper.